# A Molecular Stratification of Chilean Gastric Cancer Patients with Potential Clinical Applicability

**DOI:** 10.3390/cancers12071863

**Published:** 2020-07-10

**Authors:** Mauricio P. Pinto, Miguel Córdova-Delgado, Ignacio N. Retamal, Matías Muñoz-Medel, M. Loreto Bravo, Doris Durán, Francisco Villanueva, César Sanchez, Francisco Acevedo, Sebastián Mondaca, Erica Koch, Carolina Ibañez, Héctor Galindo, Jorge Madrid, Bruno Nervi, José Peña, Javiera Torres, Gareth I. Owen, Alejandro H. Corvalán, Ricardo Armisén, Marcelo Garrido

**Affiliations:** 1Department of Hematology & Oncology, School of Medicine, Pontificia Universidad Católica de Chile, Santiago 8330077, Chile; mauricio_pinto@outlook.com (M.P.P.); cordova.delgado.m@gmail.com (M.C.-D.); matiasm.m@outlook.com (M.M.-M.); marialoretobravo@gmail.com (M.L.B.); fco.villanu@gmail.com (F.V.); csanchez@med.puc.cl (C.S.); fnacevedo@gmail.com (F.A.); spmondaca@gmail.com (S.M.); erica.koch.hein@gmail.com (E.K.); cibanez@med.puc.cl (C.I.); hgalindo@med.puc.cl (H.G.); jmadrid@med.puc.cl (J.M.); bnervi@med.puc.cl (B.N.); penalosa@gmail.com (J.P.); gowen@bio.puc.cl (G.I.O.); corvalan@uc.cl (A.H.C.); 2Department of Physiology, Faculty of Biological Sciences, Pontificia Universidad Católica de Chile, Santiago 8331150, Chile; 3Faculty of Chemical & Pharmaceutical Sciences, Universidad de Chile, Santiago 8380494, Chile; 4Faculty of Dentistry, Universidad de los Andes, Santiago 7620001, Chile; iretamalf@gmail.com; 5Faculty of Medicine and Science, Universidad San Sebastián, Santiago 7510157, Chile; mdoris.duran@gmail.com; 6Department of Pathology, Faculty of Medicine Pontificia Universidad Católica de Chile, Santiago 8330024, Chile; javitorresm@yahoo.com; 7Advanced Center for Chronic Diseases (ACCDiS), Santiago 8330034, Chile; 8Millennium Institute on Immunology and Immunotherapy, Santiago 8331150, Chile; 9Instituto de Ciencias e Innovación en Medicina, Facultad de Medicina Clínica Alemana, Universidad del Desarrollo, Santiago 7590943, Chile; rarmisen@udd.cl

**Keywords:** gastric cancer, molecular subtypes, Epstein–Barr virus, molecular classification, targeted therapy

## Abstract

Gastric cancer (GC) is a complex and heterogeneous disease. In recent decades, The Cancer Genome Atlas (TCGA) and the Asian Cancer Research Group (ACRG) defined GC molecular subtypes. Unfortunately, these systems require high-cost and complex techniques and consequently their impact in the clinic has remained limited. Additionally, most of these studies are based on European, Asian, or North American GC cohorts. Herein, we report a molecular classification of Chilean GC patients into five subtypes, based on immunohistochemical (IHC) and in situ hybridization (ISH) methods. These were Epstein–Barr virus positive (EBV+), mismatch repair-deficient (MMR-D), epithelial to mesenchymal transition (EMT)-like, and accumulated (p53+) or undetected p53 (p53−). Given its lower costs this system has the potential for clinical applicability. Our results confirm relevant molecular alterations previously reported by TCGA and ACRG. We confirm EBV+ and MMR-D patients had the best prognosis and could be candidates for immunotherapy. Conversely, EMT-like displayed the poorest prognosis; our data suggest *FGFR2* or *KRAS* could serve as potential actionable targets for these patients. Finally, we propose a low-cost step-by-step stratification system for GC patients. To the best of our knowledge, this is the first Latin American report on a molecular classification for GC. Pending further validation, this stratification system could be implemented into the routine clinic

## 1. Introduction

Globally, gastric cancer (GC) is the fifth most common malignancy and the third leading cause of cancer death [1]. Several studies demonstrate GC is a highly heterogeneous disease [2]. This applies not only to geographic incidence and mortality rates [3,4] but also to clinical and pathological features [5]. For decades, investigators have sought to define a classification system for GC. Initial studies by Lauren classified tumors according to their histopathological and morphological features and divided GCs into intestinal or diffuse type, or mixed/undefined [6]. In general, diffuse GCs are associated to poorer prognosis and response to treatments versus intestinal type [7]. Subsequently, the World Health Organization (WHO) proposed a classification that divided GCs into five histological types: Papillary, mucinous, tubular, poorly differentiated, and signet-ring cell [8,9]. Unfortunately, these classifications are based on morphological features and therefore provide no information on potential actionable targets. Following the development of high-throughput techniques, a number of studies have proposed GC classifications based on molecular profiling rather than morphology. First, the Singapore-Duke cohort in 2011 used gene expression profiles of 248 GC patients from Singapore and Australia and divided patients into two intrinsic genomic subtypes: G-INT (genomic intestinal) and G-DIF (genomic diffuse) [10]. A subsequent study by the same group delivered a three-subtype classification, dividing patients as proliferative, metabolic, or mesenchymal subtypes [11]. More recently, The Cancer Genome Atlas (TCGA) [12] and the Asian Cancer Research Group (ACRG) [13] defined four-molecular subtype systems based on a variety of platforms; the TCGA cohort involved a total of 295 gastric adenocarcinomas and subdivided patients as Epstein–Barr virus (EBV+), microsatellite instability (MSI), chromosomal instability (CIN), and genomically stable (GS). EBV+ tumors are characterized by promoter hypermethylation, a high rate of *PIK3CA* and *ARID1A* mutations, and PDL1/2 overexpression. Similarly, MSI displays hypermethylation (particularly in *MLH1*), and a high tumor mutational burden (TMB). On the other hand, CIN tumors were enriched for *TP53* mutations, amplification of receptor tyrosine kinases (RTKs), and aneuploidy. Lastly, GS were enriched in Lauren’s diffuse-type tumors and characterized by *CDH1, RHOA*, and *ARID1A* somatic mutations. Importantly, the original TCGA report did not correlate subtype with patient prognosis; however, subsequent studies suggested they could be useful for the selection of preferred therapies. The third classification system was proposed by the ACRG and involved a total of 300 GC patients enrolled at a single institution in South Korea. Investigators defined four subtypes: MSI, microsatellite stable (MSS)-EMT, MSS-TP53+, and MSS-TP53−. Similar to TCGA, MSI tumors were hypermutated and characterized by *MLH1* loss. All other subtypes were considered MSS: First, CDH1 loss and an EMT-like signature characterized MSS-EMT. The remaining tumors were categorized according to a two-gene p53 activity signature (*CDKN1A* and *MDM2*) as MSS/TP53+ (active) or MSS/TP53− (inactive). This last subtype displayed a high rate of *TP53* mutations, aneuploidy, and *ERBB2* (HER2) amplification. Additionally, EBV+ was frequently observed in the MSS/TP53+ subtype. Unlike the TCGA, the ACRG subtypes correlated with patient survival outcomes; the worst prognosis was observed in the MSS/EMT subtype that comprises most diffuse-type tumors. In contrast, MSI displayed the best prognosis. MSS/TP53+ and MSS/TP53− had intermediate prognoses. 

Despite their outstanding contribution in the characterization of GC heterogeneity and the identification of actionable targets in the GC molecular subtypes, these stratification systems require a variety of advanced high-cost techniques in order to be implemented. Hence, their applicability in the clinic has remained limited. Herein, we report a molecular classification of Chilean GC patients using an immunohistochemistry (IHC)/in situ hybridization (ISH)-based stratification with potential applicability in the clinic. To our knowledge, this is the first Latin American report of a molecular classification for GC patients.

## 2. Results

### 2.1. A Molecular Subtype Classification for GC Based on CISH/IHC

A total of 91 GC patients were incorporated into this study. Basic clinical and pathologic characteristics are summarized in Table 1. Briefly, patients were predominantly male (66%) and advanced stage (III/IV: 64%). Most cases (88%) were gastric adenocarcinomas. First, we sought to define a molecular classification following some of the criteria defined by TCGA [12] and the ACRG [13] (shown in Figure 1a,b). Hence, we classified patients according to the subtraction algorithm shown by Figure 1c. First, patients were classified by their EBV status and subsequently by their expression of E-cadherin, mismatch repair (MMR) markers, and p53 IHC status. Initially, a total of 12 out of 91 (13%) were categorized as EBV+. Then, 15 out of 91 (17%) displayed aberrant E-cadherin expression and were classified as EMT-like. Next, we checked the expression of 4 markers of MMR status and found that 11 out of 91 (12%) were MMR deficient (MMR-D). We noticed these three subtypes displayed high exclusivity; in fact, only a single patient overlapped between subtypes being EMT-like and MMR-D. This case was finally categorized as EMT-like. Lastly, the remaining cases were subdivided as p53+ (accumulated; 19.78%) or p53− (not detected; 38.46%). Then, we tested if the GC subtypes correlated with the clinical or pathological features of the patients. Table 2 shows that the median overall survival (OS) was statistically different (*p* = 0.01) among subtypes: MMR-D patients displayed the best OS, while conversely, EMT-like patients displayed the shortest OS. Additionally, EMT-like and EBV+ patients were younger against other subtypes. The stage at diagnosis displayed differences; however, these did not reach statistical significance among subtypes (*p* = 0.058). The vast majority (93%) of EMT-like patients were advanced stage (III/IV). Conversely, in MMR-D patients, 64% were stage I/II. Human epidermal growth factor receptor-type 2 (HER2)+ and PDL1+ were also statistically different among subtypes. Interestingly, 11 out of 12 (92%) EBV+ patients were PDL1+, while in contrast only 2 out of 15 (13%) EMT-like patients were PDL1+. As expected, the p53+ subtype displayed the highest proportion of HER2+, followed by p53–. Finally, signet ring cell presence also displayed statistical differences: EBV+ tumors were all negative; on the one hand, 53% of EMT-like were positive. 

### 2.2. NGS Analysis by Subtype

Next, given the observed differences in clinical and pathology features among subtypes, we evaluated gene alterations in a panel of genes using NGS. The waterfall plot in Figure 2a shows that *TP53* was the most frequently altered across all subtypes (45%), followed by *PIK3CA* (17%) and *KRAS* (8%). The bottom panels also indicate the HER2 and PDL1 status by IHC, and TMB (mut/MB). In line with previous reports [12,13], EBV+ and MMR-D (equivalent to MSI) subtypes displayed a high frequency of *PIK3CA* mutations. Further, these subtypes along with p53+ carried a high number of genetic alterations as evidenced by their higher TMB (Figure 2a). As described in our methods, we divided p53+ (accumulated) and p53– (not detected) patients by their IHC expression. However, after analyzing our results, we noticed that some p53− cases were *TP53* mutants (Figure 2a). These included truncating and in-frame mutations. Conversely, some p53+ patients were wild type for *TP53*. Therefore, we decided to analyze a subset of patients (*n* = 41) according to their *TP53* status (Figure 2b). Interestingly, among the *TP53* wild types, we found three p53+ cases that displayed *PIK3CA* and *KRAS* mutations.

### 2.3. Differences between GC Subtypes and Patient Prognosis

As shown in Table 2, we found differences in the median OS between subtypes. Therefore, next, we sought to visualize this association using the Kaplan–Meier method (Figure 2c, *p* = 0.03). As described above, the best prognosis was observed in MMR-D patients. In contrast, EMT-like patients displayed the worst prognosis. Both p53+ and p53− displayed intermediate values and EBV+ was slightly better versus p53±. Additionally, given the discrepancy between p53 IHC and *TP53* status, we assessed survival rates by comparing *TP53* mutants versus wild type (Figure 2d). Again, subtypes displayed significant differences (*p* = 0.007); however, unlike p53±, which displayed similar survival rates, the prognosis on *TP53* mutants was markedly reduced compared to *TP53* wild types. Clinical and pathological features of *TP53* mutants and wild-type patients are shown in Appendix A.

### 2.4. Response to Chemotherapy in GC Subtypes

Current standard first-line therapy for advanced GC patients consists of a combination of fluoropyrimidines and platinum compounds [14]. Response to these chemotherapy regimens was available for a subset of 36 patients that were categorized as responders (CR or PR) or non-responders (SD or PD) by RECIST 1.1 (Appendix A). Briefly, the percentage of responders by subtype were MMR-D = 0%; EBV+ = 80%; EMT-like = 67%: *TP53*WT = 38%; and *TP53*Mutant = 63%. Unfortunately, these differences were not statistically significant (*p* = 0.31). 

### 2.5. A Low-Cost Stratification System for GC Patients

As explained, TCGA and the ACRG classification systems require a variety of high-cost techniques in order to be implemented in the routine practice. Therefore, their use in the clinic has remained limited. Here, we propose a low-cost alternative based on IHC and CISH that could be useful in the clinic. This strategy is depicted in Figure 3. In brief, patients are initially tested for HER2 and PDL1 status. Patients that stain positive should consider treatments with anti-HER2 and immunotherapy, respectively. Then, EBV infection status is detected by CISH, and EBV+ patients should consider immunotherapy. Subsequently, MMR and E-cadherin status are also determined by IHC. MMR-deficient patients should also consider immunotherapy, and patients with aberrant E-cadherin expression are classified as EMT-like. As in all subtypes, these patients should receive standard therapy but could consider the use of Claudin or KRAS (see Figure 2a) inhibitors. In this subtype, future studies should include NGS analyses in order to elucidate potential novel targets. Finally, remaining patients are classified as p53+ (accumulated) or p53− (not detected) according to IHC. In both cases, patients should receive standard therapy. They should consider combination therapies using chemotherapy plus targeted agents (such as anti-HER2). Additionally, we recommend an assessment of *TP53* status for these patients in order to obtain a more accurate prognosis.

## 3. Discussion

Like other malignancies, GC is a highly heterogeneous disease. In recent decades, a number of studies have sought to define GC molecular subtypes [10,11,12,13]. Pioneer initiatives involved a variety of platforms, including whole-exome, genome and messenger RNA sequencing, somatic copy number analysis, reverse-phase protein array profiling, and microsatellite instability (MSI) assessment. Unfortunately, given their complexity and high associated costs, the application of these stratification systems in the everyday clinic is neither feasible nor cost effective, and therefore, their impact has remained limited. Still, a couple of studies have sought to apply stratification into molecular subtypes based on IHC and ISH methods. Indeed, a retrospective study by Setia et al. [15] classified a group of 146 North American gastric adenocarcinoma patients according to the expression levels of a set of 14 biomarkers. Similar to our report, the investigators proposed a five-tier classification into molecular subtypes: EBV+, MSI, aberrant E-cadherin, aberrant p53, and normal p53. In line with our findings, this study found that EBV+ and MSI (inferred from MMR-D) patients had better survival rates. Unfortunately, the clinical and pathological characteristics of patients in this study were limited to age at diagnosis, male/female ratio, and tumor location. A second study used a similar approach with 5 clusters in a cohort of 349 gastric adenocarcinomas from a single institution in South Korea [16]. This study further confirmed EBV+ and MSI tumors as those with the best OS. In contrast, aberrant E-cadherin-subtype (equivalent to EMT-like or GS) tumors were associated with the poorest survival rates. More recently, a subsequent study by the same group expanded these findings by adding a validation cohort, reporting a total of 894 consecutive patients, confirming an association between the molecular subtype and OS [17]. Interestingly, the authors compared their results against the abovementioned study by Setia et al. [15] and found that the distribution of subtypes in Asian or Western cohorts displayed important differences. Notably, the distribution of subtypes in our cohort also differed from the abovementioned Asian and North American studies (see Appendix A). In particular, the percentage of EBV+ in our study is markedly higher (13%) versus the abovementioned studies (5%, 7%, or 9%, respectively). However, this is a characteristic of the Chilean population that has been reported previously [18,19]. Regarding the percentage of the MMR-D subtype (12%), this is fairly close to the 14% commonly observed in the routine clinic and reported by similar studies [20]. In contrast, TCGA and ACRG reported a much higher percentage of MSI: 21.7% and 22.7%, respectively. Evidently, this could be explained by methodological differences. Our study reports MMR status; as explained above, although MSI status can be inferred from MMR deficiency, these do not completely overlap. Similarly, the observed differences in the percentages of p53± (Appendix A) can also be partially attributed to methodology; cutoff levels and criteria for p53 positivity/negativity are somehow inconsistent among studies. Indeed, the study by Setia et al. [15] establishes p53+ (i.e., aberrant) cases as those with a diffuse strong expression or those with a complete loss of signal. In contrast, a weak patchy signal is categorized as p53– (normal). The study by Ahn et al. [16] applies similar criteria classifying diffuse/strong or complete loss of p53 as aberrant (p53+). Then, a subsequent study established a ≥10% cutoff of a strong nuclear-positive stain in tumor cells for p53+. Cases that were <10% or those with a weak scattered or patchy expression were classified as p53− [17]. Although initially we applied a similar criteria and a 10% cutoff for p53+, we eventually opted for a more stringent cutoff value set at 20% that reduced the number of false positives, meaning patients classified as aberrant (p53+) but having a wild type *TP53* gene from 12 to 3 (see comparison in Appendix A). However, despite these methodological variations, our study confirms a number of previous reports that indicate geographical differences in GC epidemiology, and clinical and molecular features [21,22,23,24]. Importantly, to the best of our knowledge, our study is the first report on a GC molecular subtype classification with potential clinical applicability in Latin America. Our study confirms that EBV+ and MSI are associated with a better prognosis. Additionally, EMT-like displayed the poorest prognosis. 

As pointed out, current standard first-line treatment for advanced GC consists of combined chemotherapy regimens. However, only a few studies have evaluated the chemotherapy response by GC molecular subtypes. A recent study by Kubota et al. [25] reported shorter progression-free survival (PFS) in MMR-D patients treated with standard first-line chemotherapy. Subsequent addition of anti-PD1 therapy improved outcomes (both OS and PFS) for these patients. In contrast, studies have reported a favorable response to first-line chemotherapy in both localized [26] and advanced [27] disease EBV+ cases. Furthermore, clinical studies suggest the addition of immune checkpoint inhibitors to first-line chemotherapy in metastatic EBV+ patients may increase the chances of a durable or complete response [27]. On the other hand, adjuvant chemotherapy has demonstrated significant benefits for stage II/III p53+ or EMT-like patients versus other subtypes [28]. Our preliminary data confirm these findings (Appendix A); the percentage of responders (CR or PR) was the highest on EBV+ (4 out of 5; 80%) followed by EMT-like (6 out of 9; 67%). Conversely, our two MMR-D cases were non-responders. 

Evidently, the ultimate goal of molecular stratification systems is to divide tumors into subtypes enriched in actionable targets for personalized therapies. In this regard, our study confirmed most EBV+ tumors were PDL1+ (Figure 2a), and therefore these are candidates for immunotherapy [29]. Similarly, it is well documented that MSI tumors, characterized by MMR deficiency, display a favorable response to checkpoint inhibitor-based immunotherapy [30]. Unlike EBV+ or MSI, potential targeted therapies for other subtypes are still undefined. However, in line with previous reports, our data show that the p53+ subtype (equivalent to CIN or MSS/TP53−) is enriched for HER2 amplification/alterations [13]; therefore, a subset of these patients could be candidates for Trastuzumab or anti-HER2 therapies. Indeed, following the results of the ToGA trial, current guidelines include a mandatory HER2 IHC for GC patients [31]. As described previously, this subtype also displays amplification of other receptor tyrosine kinases (RTKs) [15,16], such as fibroblast growth factor receptor-type 2 (FGFR2) (Figure 2a). Recent studies demonstrate the efficacy of FGFR inhibitors in cholangiocarcinoma patients [32] and could be an option for p53+/anti-HER2 refractory patients; however, this should be further evaluated in clinical trials. To date, the EMT-like subtype (equivalent to GS or MSS/EMT) remains the most challenging in terms of targeted therapies. Our results indicate a subset of these tumors displayed potentially actionable alterations, such as PDL1+ or *FGFR2/KRAS* gene amplifications (Figure 2a). Indeed, previous studies suggest GS tumors are characterized by *ARID1A* mutations [33] and these alterations correlate with higher PDL1 expression, suggesting that these patients could be candidates for immunotherapy [34]. As pointed out above, FGFR2 may also be an alternative for a subset of EMT-like patients. Regarding *KRAS* amplification, a study sought to correlate *KRAS* status against GC histological phenotypes and found *KRAS* amplification was associated to the poorly differentiated solid type [35], which is a characteristic feature of diffuse GCs. Furthermore, *KRAS* activation stimulates EMT and promotes metastasis in GC preclinical models [36]. Although they are yet to be tested in GC, KRAS inhibitors have been used successfully in lung and colon cancer patients [37]. 

Based on our findings and the existing literature, we propose a low-cost stratification system for advanced GCs into molecular subtypes (Figure 3). This step-by-step guide allows classification of tumors into five subtypes. In all cases, we recommend the use of standard GC therapy; however, we provide therapeutic alternatives that might be considered for each subtype and recommendations for future clinical studies. Even though this stratification could be useful in order to develop more personalized treatments, there are several issues that should be resolved before such a strategy can be fully implemented into the clinic. First, as reported by others [16], our study used IHC data obtained from a TMA [38,39]. Although this is a reliable and validated method to assess protein expression, it involves only a small section of the tumor and therefore it may not account for the intra-patient biomarker heterogeneity observed in GC [40] and therefore this is a limitation of our study. The use of circulating tumor cells or circulating tumor DNA obtained from liquid biopsy (blood) samples might offer an alternative to solve this issue [41]; however, these assays should be further validated in the clinical setting. A second issue is related to the use of protein expression, which may be inadequate to assess the genomic complexity behind molecular alterations. Our *TP53* data (Figure 2b) is a case in point, as some *TP53*WT tumors were categorized as accumulated (or aberrant) p53+ by IHC, or vice versa. Interestingly, some *TP53*WT tumors classified as p53+ displayed *MDM2* amplification (Figure 2b). This is a p53-specific ubiquitin E3-ligase [42] that targets p53 for degradation. A similar issue might occur with our aberrant E-cadherin subtype (EMT-like). In this case, we assumed *CDH1* alterations; however, this relationship is still controversial [43]. Again, this is another limitation of our study and future studies should refine the panel of specific biomarkers for a more accurate stratification. Finally, this was a retrospective study with a relatively low number of patients and therefore our findings should be further confirmed by a larger cohort in prospective studies. 

## 4. Materials and Methods 

### 4.1. Patients

A total of 91 GC patients were retrospectively included into our study. All patients were recruited at the Cancer Center in the Red Salud UC-Christus and Pontificia Universidad Catolica de Chile. Patient inclusion criteria were age ≥18, diagnosed with confirmed GC, with ≥3 months of clinical follow-up, and able and willing to sign an informed consent. A subset of these patients were previously characterized as part of the FORCE1 study [19]. All participants signed a consent form to publish. A waiver of consent was granted to include deceased patients in the study. The Internal Review Board and the Ethics and Scientific Committee at the School of Medicine approved this research. This study was conducted in accordance with the principles of the Declaration of Helsinki.

### 4.2. Hematoxylin and Eosin and Immunohistochemistry (IHC) Assays

For analyses, paraffin blocks were obtained, cut, and stained by hematoxylin and eosin (H&E) in order to select the best histological area. Subsequently, selected tissue areas were placed into the TMA by circling the identified area in the corresponding block. Cylindrical core biopsies were then extracted from each paraffin block using a 20-μm^2^ stylet and placed into a new recipient block. Selected adequate cases had tumors that occupied at least 10% of the core area. Each case was processed in triplicate to prevent tissue loss during cutting. Sections from each tissue array block were cut, de-paraffinized, and dehydrated for H&E and immunohistochemical procedures. Our IHC analyses included the following primary antibodies: PD-L1: Cat # SK00521-k (Dako, Carpinteria, CA, USA); all other antibodies: E-cadherin: Cat # 7904497 MLH1: Cat # 06472966001; PMS2: Cat # 06419216001; MSH2: Cat # 05269270001; MSH6: Cat # 5929911001); HER2: Cat # 05278368001; p16: Cat # 06695221001; p53: Cat # 5278074001 were from Roche Diagnostics (Basel, Switzerland). Protein expression levels were determined in a manually prepared TMA [38,39] from deparaffinized sections obtained from archival tumor samples.

### 4.3. IHC for HER2 and PDL1

HER2+ status was assessed following a protocol described previously [44]. Briefly, HER2+++ by IHC analysis was considered positive, and HER2++ cases were further confirmed by fluorescence in situ hybridization (FISH). Samples that were + or 0 by IHC were considered negative. PDL1 status was determined from the combined positive score (CPS); this score assesses the proportion of PDL1+ tumor cells and PDL1+ tumor-associated cells divided by the total number of cells × 100. In our study, we set a cutoff of CPS ≥10 for PDL1+ patients following the recommendation of the upcoming KEYNOTE062 study (ClinicalTrials.gov identifier: NCT02494583).

### 4.4. Epstein–Barr Virus (EBV) and Mismatch Repair (MMR) Status Determination

EBV status was determined using paraffin block sections obtained from FFPE samples. EBER1 expression was detected by a chromogenic in situ hybridization (CISH) method described previously [18,19]. MMR status was determined using a set of 4 biomarkers assessed by IHC [19]. These are DNA repair enzymes coded by the MLH1, PMS2, MSH2, and MSH6 genes. As described above, these were evaluated by TMA. Patients were then classified as intact or loss for each assessed marker using specific antibodies. Those that displayed loss in any of these 4 mismatch-repair enzymes were then classified as MMR deficient (MMR-D). Otherwise, if all 4 markers were found to be intact, the patient was considered MMR proficient.

### 4.5. E-Cadherin and p53 Status Determination

E-cadherin is normally expressed in epithelial cells’ membranes [45]. Samples with a strong membrane staining were considered normal. In contrast, samples that displayed partial/complete loss of membrane staining or cytoplasmic staining pattern were classified as aberrant. Studies have demonstrated that p53 is a DNA repair protein with a high turnover rate [46]; therefore, its accumulation is indicative of alteration. Hence, p53 expression status was considered either as accumulated (p53+) or not detected (p53−). Histologically, p53+ was defined by strong nuclear staining in at least 20% of tumor cells. Otherwise, if p53 staining was weak, patchy, scattered, completely lost, or observed in <20% of cells, the sample was considered p53−.

### 4.6. Genomic Analysis

Nucleic acids were extracted from FFPE samples using the RecoverAll kit (Cat #AM1975, Thermo Fisher, (Carlsbad, CA, USA) and were analyzed with a Oncomine comprehensive assay v.1 kit (Thermo Fisher, Carlsbad, CA, USA) [47]. Annotation of detected alterations was performed by Oncomine Variant Annotator plugin 2.2.7 (OVA), supported by the dbSNP, ClinVar, and VariantDB databases, following the manufacturer’s suggestions. Tumor mutational burden (TMB) was estimated from the number of somatic non-synonymous variants per Megabase (mt/Mb) identified at the hotspot regions interrogated by the Oncomine Comprehensive v1 panel. 

### 4.7. Outcomes and Statistical Analysis

Comparisons of categorical variables between subgroups were tested using Fisher’s exact and continuous variables were compared with ANOVA. Overall survival (OS) was defined as the time lapse between cancer diagnosis and death by GC (event), or the last follow-up visit (censorship), or the death by any other cause (censorship). Descriptive survival analysis was performed using the Kaplan–Meier method and the differences between molecular subtypes were estimated using the cox-trend test. Tumor response was assessed in patients with measurable lesions using the Response Evaluation Criteria in Solid Tumors version 1.1 identifying patients with partial response (PR), complete response (CR), progressive disease (PD), or stable disease (SD). Statistical analyses were performed using STATA (StataCorp. 2017. Stata Statistical Software: Release 15. College Station, TX, USA) or R software (The R Foundation, Vienna, Austria). Statistical significance was set at *p* < 0.05.

## 5. Conclusions

In summary, our study confirmed that an IHC/CISH-based stratification of GC could be a cost-effective and clinically useful tool in routine practice. However, more validation studies are still required. Perhaps the main obstacle for the implementation is the small size and availability of diagnostic endoscopic biopsy material that may not be representative of the heterogeneity of the entire tumor. To the best of our knowledge, our report is the first of its kind in Latin America, delivering a molecular subtype stratification and NGS analysis confirming common gene alterations and actionable targets within subtypes. Future studies should further validate and expand our findings within Latin America.

## Figures and Tables

**Figure 1 cancers-12-01863-f001:**
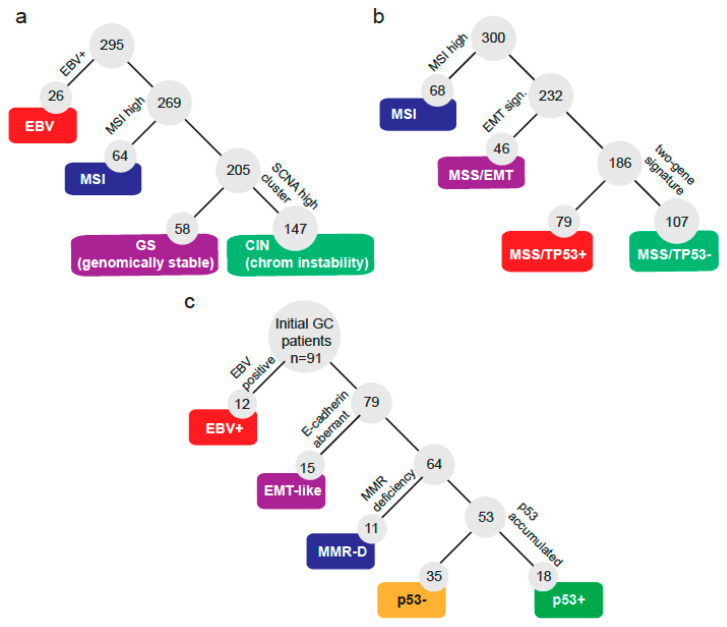
Molecular classification systems for gastric cancers. Subtraction algorithms and molecular subtypes defined by (**a**) The Cancer Genome Atlas, (**b**) the Asian Cancer Research Group, or (**c**) by the current study.

**Figure 2 cancers-12-01863-f002:**
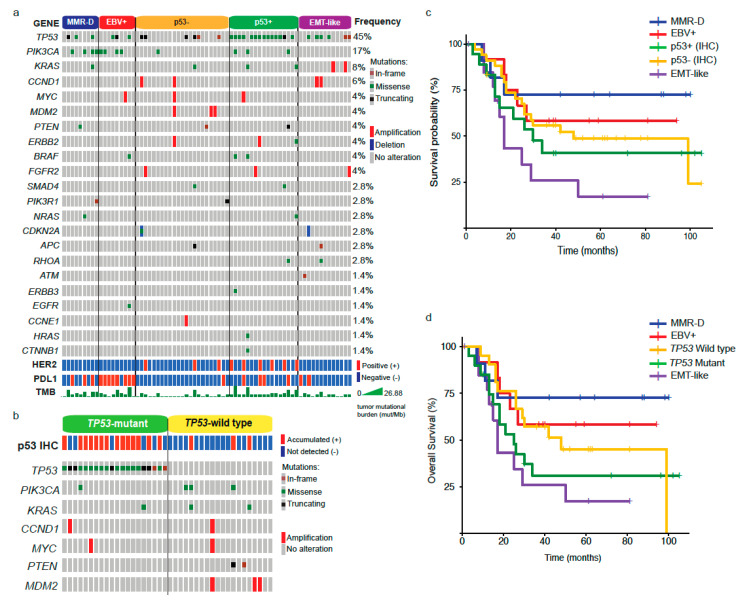
Molecular alterations and prognosis by GC subtypes. (**a**) Next Generation Sequencing (NGS) analysis, HER2, PDL1 status, and tumor mutational burden by GC subtype. (**b**) Genomic alterations and p53 status in *TP53* mutants versus *TP53* wild type. (**c**) Kaplan–Meier plot showing the overall survival by GC subtypes. (**d**) Kaplan–Meier plot comparing OS between *TP53* mutants and wild type.

**Figure 3 cancers-12-01863-f003:**
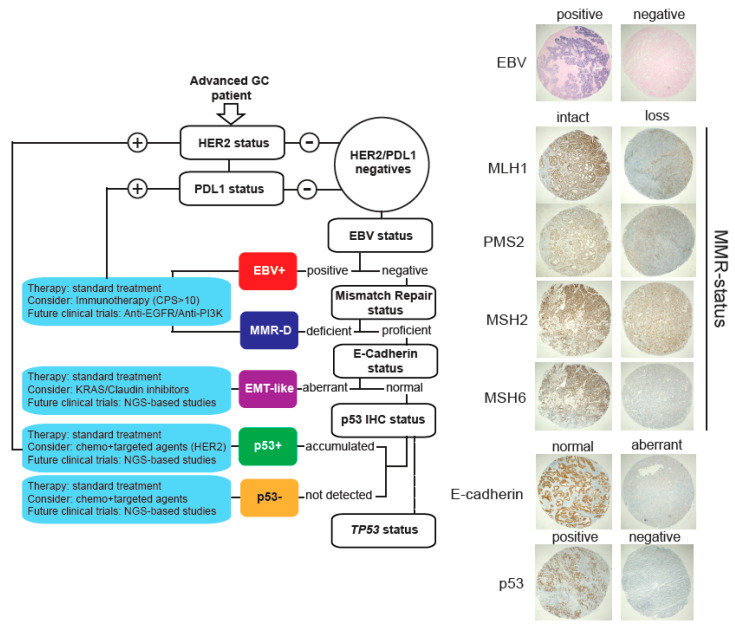
Proposed molecular stratification system for advanced gastric cancer patients. Briefly, our proposal includes assessment of HER2 and PDL1 status by IHC. Then, patients are stratified according to EBV positivity, MMR, E-cadherin, and p53 status. Whenever possible, we also recommend *TP53* analysis. Right panels show representative images for each subtype.

**Table 1 cancers-12-01863-t001:** Basic clinico-pathological characteristics of the patient cohort.

Characteristic	Units (IQR or %)
Median OS; months	30 (16–61)
Males	60 (66)
Females	31 (34)
Median age; years	63 (55–73)
Stage	
I/II	33 (36)
III	46 (51)
IV	12 (13)
Lauren histotype	
Intestinal	27 (37)
Diffuse	31 (42)
Mixed	15 (21)
WHO classification	
Adenocarcinoma	77 (88)
Undifferentiated carcinoma	8 (9)
Adeno-squamous carcinoma	3 (3)
Primary tumor location	
Proximal	26 (30)
Medial	30 (34)
Distal	27 (31)
Multiple	4 (5)
Signet ring cells	
None	58 (64)
<50%	27 (30)
≥50%	6 (6)
HER2+ status	12 (13)
PDL1+ status	26 (29)
EBV+ status	12 (13)
MMR-deficient	13 (14)
p53+ status	39 (43)
E-cadherin loss	15 (17)
Comorbidities	
None	46 (51)
1	20 (22)
2	18 (20)
≥3	7 (8)

**Table 2 cancers-12-01863-t002:** Clinical and pathological features by gastric cancer subtype.

Characteristic	MMR-D *n* = 11	EBV+ *n* = 12	p53+ *n* = 18	p53– *n* = 35	EMT-Like *n* = 15	Total *n* = 91	*p*-Value
Median OS; Months (range)	64 (17–98)	38 (20.5–57)	28 (13–40)	39.5 (18–62)	16 (12–29)	30 (16–61)	0.01
Males; *n* (%)	6 (55)	11 (92)	11 (61)	22 (63)	10 (67)	60 (66)	0.31
Median age; years (IQR ^1^)	66 (64–70)	57 (52–69)	63.5 (56–71)	64.5 (55–76)	57 (50–75)	63.5 (55–73)	0.22
Stage; *n* (%)							
I/II	7 (64)	6 (50)	9 (50)	10 (29)	1 (7)	33 (36)	0.058
III	4 (36)	4 (33)	7 (39)	20 (57)	11 (73)	46 (51)	
IV	0	2 (17)	2 (11)	5 (14)	3 (20)	12 (13)	
Primary tumor; *n* (%)							
Proximal	1 (9)	9 (75)	5 (29)	8 (25)	3 (20)	26 (30)	0.081
Medial	3 (27)	2 (17)	5 (29)	14 (44)	6 (40)	30 (34)	
Distal	7 (64)	1 (8)	6 (35)	8 (25)	5 (33)	27 (31)	
Multiple	0	0	1 (6)	2 (6)	1 (7)	4 (5)	
IHC ^2^; *n* (%)							
HER2+	1 (9)	0	6 (33)	5 (14)	0	12 (13)	0.037
PDL1+	3 (27)	11 (92)	5 (28)	5 (14)	2 (13)	26 (29)	<0.001
Signet ring cells; *n* (%)							
No	7 (64)	12 (100)	14 (78)	18 (51)	7 (47)	58 (64)	0.017
<50%	4 (36)	0	3 (17)	12 (34)	8 (53)	27 (30)	
≥50%	0	0	1 (6)	5 (14)	0	6 (7)	
Lauren histotype; *n* (%)							
Diffuse	5 (56)	2 (29)	3 (20)	14 (50)	7 (50)	31 (42)	0.038
Intestinal	2 (22)	4 (57)	10 (67)	10 (36)	1 (7)	29 (40)	
Mixed	2 (22)	1 (14)	2 (13)	4 (14)	6 (43)	13 (18)	

^1^*IQR* interquartile range, ^2^
*IHC* immunohistochemistry.

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
