# Peer review of "A Molecular Stratification of Chilean Gastric Cancer Patients with Potential Clinical Applicability"

_cancers, 2020, doi:10.3390/cancers12071863_

Round 1
Reviewer 1 Report
I recently received for evaluation the paper "A molecular stratification of Chilean gastric cancer patients with potential clinical applicability" by Pinto et al.
I appreciated reading your paper and I believe that a few changes must be done before the article can be considered for publication:
- The small number of patients does reduce the impact of your findings. This is particularly true when you consider that patients enrolled in the analysis have widely different stages at diagnosis and that only a few of them have actually received chemotherapy (32/91, 35%). This is the strongest limitation to the work as it cannot help clinicians in deciding whether different treatment options should be offered on the basis of your classification. You should either increase the number of patients that have actually received chemotherapy and submit them to testing or either have a relatively more homogeneous patient population.
- From what I can see from Table 2 there is a statistically significant difference in stages among the different classes. This should be taken into account into some form of multivariate analysis (that is lacking) because, as it stands, one cannot tell whether the difference in overall survival is due to different stage at diagnosis rather than by your molecular/histological classification.
- I appreciated the methods section because it is very well written and describes in detail how you performed the experiment: I think that it helps also others to reproduce your findings. I have only a few comments on that: the percentage of patients that are included in the D-MMR subgroup (17% in your analysis) is actually more similar to what is actually seen in everyday clinical practice (PMID:26796888, DOI:10.1007/s10120-016-0594-4 ) more than from what is usually described in both TCGA and ACRG classification. Part of this difference might be due to different evaluation method (D-MMR/P-MMR status is not exactly overlapping to MSI-H/MSS status). I would like to see a comment about that in the discussion.
- Looking at p53 accumulated/not detected vs TP53 mutated/not mutated analysis it seems that they are quite different also in terms of survival outcomes. It is not described anywhere in the text if there is concordance between the 2 evaluations (and it should be calculated and added). Furthermore, don't you think that this might have been due to the fact that you used 10% cut-off value for IHC for p53? There are actually other published papers that use 15-20% cut-off value for IHC for p53. Indeed, by looking at survival curves while TP53 mutational status does determine significant differences in survival, the difference in survival between IHC p53+ and p53- patients is rather small. This might have been due to having used a rather "low" cut-off value to estabilish p53 mutated status. I would suggest trying with a higher cut-off treshold as I think that it would increase reproducibility (and concordance with NGS mutational TP53 analysis).
Actions- Search in PubMed
- Search in NLM Catalog
- Add to SeaThe methods section is very accurate and descriptive and I appreciated a lot this part of the work (as it allows to reproduce your results also by other authors). I have only a few comments: your detection rate for D-MMR patients (about 10%) is actually more similar to what is usually seen in patients with more advanced stages of involvement (like in our previous work,Albeit your classification is focused on assessing differences in overall survival among the 5 classes that you identified, you did not take into account other prognostic factors that are crucial for survival: in particular, from what I can see from Table 2, there is an obvious (and statistically significant) difference in stages among the 5 classes (most patients in D-MMR subgroup for example are in earlier stages vs those with p53 mutated or EMT subgroup that are either in stage III or IV). This would have been partly corrected by some form of multivariate analysis that would have taken into account stage and the classes that you identified but it was not performed. Multivariate analysis must be performed because, as it stands now, one cannot tell whether differences in survival are due to stage unbalancement or histological/molecular subtype.
Author Response
We would like to thank the reviewer for his/her kind comments. We have prepared point-by-point replies to the comments and suggestions below
REVIEWER 1
I recently received for evaluation the paper "A molecular stratification of Chilean gastric cancer patients with potential clinical applicability" by Pinto et al.
I appreciated reading your paper and I believe that a few changes must be done before the article can be considered for publication:
1.The small number of patients does reduce the impact of your findings. This is particularly true when you consider that patients enrolled in the analysis have widely different stages at diagnosis and that only a few of them have actually received chemotherapy (32/91, 35%). This is the strongest limitation to the work as it cannot help clinicians in deciding whether different treatment options should be offered on the basis of your classification. You should either increase the number of patients that have actually received chemotherapy and submit them to testing or either have a relatively more homogeneous patient population.
R1 Thank you for reviewing our work. As the reviewer indicates this is perhaps the main limitation of our study and we acknowledge this in our manuscript. However, since this is the first report of a molecular classification with potential clinical utility in Latin America we feel it is worth sharing our results. Gastric cancer is the leading cause of cancer death in Chile and comparing our population (17.5 M) against South Korea (51.6 M) or the United States (328.2 M) our numbers represent a significant proportion of patients. Unfortunately, given the retrospective nature of our study we cannot add patients into this report. As the reviewer suggests, our research group is currently working in a prospective study in order to validate and expand these data.
2.From what I can see from Table 2 there is a statistically significant difference in stages among the different classes. This should be taken into account into some form of multivariate analysis (that is lacking) because, as it stands, one cannot tell whether the difference in overall survival is due to different stage at diagnosis rather than by your molecular/histological classification.
R2.Please see our reply to comment 5 below (R5) that also involves the response to this comment since both are directly related.
3.I appreciated the methods section because it is very well written and describes in detail how you performed the experiment: I think that it helps also others to reproduce your findings. I have only a few comments on that: the percentage of patients that are included in the D-MMR subgroup (17% in your analysis) is actually more similar to what is actually seen in everyday clinical practice (PMID:26796888, DOI:10.1007/s10120-016-0594-4 ) more than from what is usually described in both TCGA and ACRG classification. Part of this difference might be due to different evaluation method (D-MMR/P-MMR status is not exactly overlapping to MSI-H/MSS status). I would like to see a comment about that in the discussion.
R3.Thank you for your kind comments. In response to your suggestions we have added the recommended article into our revised manuscript (new reference #20). In addition, as requested we have added a short statement in the discussion section explaining the difference between MSI and MMR-D. Starting at LINE 349
4.Looking at p53 accumulated/not detected vs TP53 mutated/not mutated analysis it seems that they are quite different also in terms of survival outcomes. It is not described anywhere in the text if there is concordance between the 2 evaluations (and it should be calculated and added). Furthermore, don't you think that this might have been due to the fact that you used 10% cut-off value for IHC for p53? There are actually other published papers that use 15-20% cut-off value for IHC for p53. Indeed, by looking at survival curves while TP53 mutational status does determine significant differences in survival, the difference in survival between IHC p53+ and p53- patients is rather small. This might have been due to having used a rather "low" cut-off value to establish p53 mutated status. I would suggest trying with a higher cut-off threshold as I think that it would increase reproducibility (and concordance with NGS mutational TP53 analysis).
R4.This is a very interesting point. As the reviewer points out cutoff values and criteria for p53 positivity/negativity for GC are inconsistent among publications. Following the recommendation by the reviewer we increased our cutoff value from 10% to 20% and reanalyzed all of our data. Consequently, figures 1 and 2 and table 2 and supplementary FigS1 have been modified compared to our original submission. Indeed, this new cutoff value reduced the number of false positives in our study from 12 to 3. These changes have been incorporated in the revised version of our manuscript starting at LINE 354. Interestingly, all 3 studies that have postulated the use of IHC/ISH based molecular subtypes have applied different criteria, which could evidently affect the % of subtypes reported by different studies. This is also now incorporated in the revised discussion section of our manuscript. Also, to further clarify this point we have added a new supplementary figure S2 that shows the number of false positives or negatives using different criteria for p53.
5.The methods section is very accurate and descriptive and I appreciated a lot this part of the work (as it allows to reproduce your results also by other authors). I have only a few comments: your detection rate for D-MMR patients (about 10%) is actually more similar to what is usually seen in patients with more advanced stages of involvement (like in our previous work, Albeit your classification is focused on assessing differences in overall survival among the 5 classes that you identified, you did not take into account other prognostic factors that are crucial for survival: in particular, from what I can see from Table 2, there is an obvious (and statistically significant) difference in stages among the 5 classes (most patients in D-MMR subgroup for example are in earlier stages vs those with p53 mutated or EMT subgroup that are either in stage III or IV). This would have been partly corrected by some form of multivariate analysis that would have taken into account stage and the classes that you identified but it was not performed. Multivariate analysis must be performed because, as it stands now, one cannot tell whether differences in survival are due to stage unbalancement or histological/molecular subtype.
R5.We agree; and this is definitely another limitation of our study. However there are some considerations in this point. First, according to the type of data we collected the most appropriate multivariate model would be a time-to-event regression model such as the Cox model. However, given the small number of patients (another limitation we acknowledge in our manuscript) this is not feasible. As the reviewer points out differences in OS could be due to stage at diagnosis (64% are stage III or IV). Unfortunately, since we do not have enough “events per predictor parameters” 1(see reference below) we do not have the required statistical power to reliably test this hypothesis. In fact, some studies have postulated a well-used “rule of thumb” for sample size that should include at least 10 events per candidate predictor (variable) 2. As you can see some of proposed subtypes (EBV+ and MMR-D) do not reach that number (see table). Consequently, we are currently in a national prospective study seeking to validate and expand our results in a larger cohort and establish if our subtypes are independently associated with OS or if this is determined by other factors such as stage or age at diagnosis. We hope this is acceptable for the reviewer
REF1 Riley, R. D. et al. Minimum sample size for developing a multivariable prediction model: PART II - binary and time-to-event outcomes. Stat. Med. 38, 1276–1296 (2019).
REF2 Concato, J., Peduzzi, P., Holford, T. R. & Feinstein, A. R. Importance of events per independent variable in proportional hazards analysis. I. Background, goals, and general strategy. J. Clin. Epidemiol. 48, 1495–1501 (1995).

Reviewer 2 Report
this is a very interesting study validating gastric cancer subtyping by IHC and ISH in a South American cohort of patient. the study is well designed and I only have a request for the authors:
Representative images should be shown in the paper, especially highlighting the different subtypes. a more thorough description on the methods for IHC/ISH and how samples are prepared, imaged will be needed. also images representing different scores should be shown. As the patient set is very unique and well balanced, it would be great if the images/data are being made available by teh authors in public databases
Author Response
We would like to thenk you for reviewing our work. Below we have prepared point-by-point replies to your comments and suggestions
REVIEWER 2
This is a very interesting study validating gastric cancer subtyping by IHC and ISH in a South American cohort of patient. the study is well designed and I only have a request for the authors:
Thank you for your comments and for reviewing our work
1.Representative images should be shown in the paper, especially highlighting the different subtypes. a more thorough description on the methods for IHC/ISH and how samples are prepared, imaged will be needed. also images representing different scores should be shown. As the patient set is very unique and well balanced, it would be great if the images/data are being made available by teh authors in public databases
R1.Correct. In response to the request by the reviewer we have added more details regarding preparation of samples and IHC procedures in the methods section. Also, we have added representative images into the new revised figure 3 of our manuscript. Finally, we are also adding our complete datasets in this new submissions to make our data available to the general public

Round 2
Reviewer 1 Report
I received and reviewed the manuscript "A molecular stratification of Chilean gastric cancer patients with potential clinical applicability" after the authors have performed changes to the work as my previous request.
Authors have made several changes to the work and I have appreciated mostly the fact that they complied to my requests. I have appreciated that, after revision of the work, results of the analysis have been improved and I think that in the current form the paper might suggest a easy method that has clinical applicability for molecular stratification of gastric cancer patients also in the real world setting.
I am convinced that in the current form the article is suitable for publication.
Author Response
Thank you for taking the time to review our work. We also believe this new version has improved from its original version thanks to your comments and suggestions